# Effects of *Allium hookeri* Extracts on Hair-Inductive and Anti-Oxidative Properties in Human Dermal Papilla Cells

**DOI:** 10.3390/plants12091919

**Published:** 2023-05-08

**Authors:** Seokmuk Park, Nayeon Han, Jung-Min Lee, Jae-Ho Lee, Seunghee Bae

**Affiliations:** 1Department of Cosmetics Engineering, Konkuk University, 120 Neungdong-ro, Gwangjin-gu, Seoul 05029, Republic of Korea; ted968@konkuk.ac.kr (S.P.); nina1305@konkuk.ac.kr (N.H.); jaeho3909@hanmail.net (J.-H.L.); 2Derma Bio Medical Research Center, Dermato Bio, Inc., 174-1 Songdo-dong, Yeonsu-gu, Incheon 21984, Republic of Korea; jungmin-lee@skinbutak.com

**Keywords:** alopecia, anti-hair loss, anti-oxidant, dermal papilla, hair inductivity, *Allium hookeri*, apoptosis, senescence

## Abstract

Oxidative stress and cellular senescence in dermal papilla cells (DPCs) are major etiological factors causing hair loss. In this study, the effect of the *Allium hookeri* extract (AHE) on hair-inductive and anti-oxidative properties was investigated in human DPCs. As a result, it was found that a non-cytotoxic concentration of the extracts increased the viability and size of the human DPC spheroid, which was associated with the increased expression of hair-growth-related genes in cells. To determine whether or not these effects could be attributed to intracellular anti-oxidative effects, liquid chromatography-mass spectrometry alongside various biochemical analyses are conducted herein. An ingredient called alliin was identified as one of the main components. Furthermore, AHE treatment induced a significant decrease in H_2_O_2_-mediated cytotoxicities, cell death, and cellular senescence in human DPCs. Upon analyzing these results with a molecular mechanism approach, it was shown that AHE treatment increased β-Catenin and NRF2 translocation into the nucleus while inhibiting the translocation of NF-κB (p50) through p38 and PKA-mediated phosphorylations of GSK3β, an upstream regulator of those proteins. These results overall indicate the possibility that AHE can regulate GSK3β-mediated β-Catenin, NRF2, and NF-κB signaling to enhance hair-inductive properties and ultimately protect against oxidative stress-induced cellular damage in human DPCs.

## 1. Introduction

Hair follicles (HFs) are a visible feature of mammals and are complex mini-organs that have functions involving thermoregulation, sensory activity, physical protection, and social interactions [1]. HFs are tissues composed of many cell clusters, with signal transmission between epidermal and dermal compartments playing a key role in HF regeneration [2]. Among them, dermal papilla cells (DPCs), which are specialized fibroblasts of mesenchymal cells, are indispensable for initiating HF regeneration via providing instructive signals including FGF7, FGF10, TGF-β2, and noggin [3,4]. Wingless (WNT) signals facilitate the release of these secreted proteins in the DPCs, whilst paracrine FGF7 or FGF10 signaling by DPCs stimulates the proliferation of the hair germ [4,5]. Wnt/β-catenin signaling plays a significant role in both the telogen-to-anagen transition and HF morphogenesis [5,6,7]. The absence of β-catenin prevents HF regeneration [8], whereas the upregulation of β-catenin activity in DPCs increases the length of HFs as well as the number of proliferating DPCs in each anagen follicle, thus emphasizing the importance of Wnt/β-catenin signaling in hair growth [5,9,10].

The miniaturization of HFs and vellus hairs is known as one of the major clinical presentations of alopecia [11,12,13]. Although the pathogenesis and precise molecular mechanism of hair loss remain largely unknown, several recent studies have compellingly suggested that abnormal androgen metabolism and androgen receptor (AR)-mediated signaling can induce the regression of the hair cycle, eventually leading to hair loss [14,15,16,17,18,19,20]. Additionally, DPCs from the balding scalps of patients with androgenetic alopecia (AGA) exhibited senescent-like phenotypes characterized by a morphology that includes flat and enlarged cells, SA-β-gal staining, and the increased expression of cyclin-dependent kinase inhibitors (CDKIs) [21,22,23,24].

Cellular senescence was shown to occur due to many different triggers, whilst senescent cell accumulation induced tissue dysfunction and organismal aging [25]. Skin aging occurs alongside hair loss, owing to defects in hair follicle epithelial and mesenchymal niches [26]. Recent studies have also shown that DPCs undergo progressive cell loss with aging, eventually being miniaturized, which contributes to the pathogenesis of HFs. Furthermore, with aging, hair follicle dermal stem cells (hfDSCs), which replenish DPCs in niches, become senescent-like phenotypes and senescence-associated secretory phenotypes (SASP) [26,27]. These results suggest that the intrinsic and extrinsic aging of HFs is closely associated with hair loss.

Oxidative stress is one of the major triggers of aging [28]. Various cellular processes produce reactive oxygen species (ROS) with ROS-mediated oxidative stress promoting DNA damage, protein degradation, peroxidative damage in lipids, and the inhibition of repair. All these intracellular reactions can then cause cellular senescence [29]. Various studies have previously demonstrated that oxidative stress levels are significantly increased with aging or when undergoing the pathogenesis of alopecia [21,30,31,32,33]. DPCs from balding scalps were found to be far more vulnerable to oxidative stress than DPCs from non-balding scalps were, whilst in response to H_2_O_2_, they released more negative hair growth regulators [22]. Furthermore, H_2_O_2_ induced premature catagen and hair growth inhibition, whereas Nrf2 activation significantly ameliorated H_2_O_2_-dependent hair growth inhibition [34]. These findings suggest that ROS may play a key role in hair loss through the modulation of senescence, apoptosis, and hair growth-regulatory genes in DPCs, in an androgen-independent manner. Therefore, several studies have been focused on identifying therapeutic candidates using DPCs with H_2_O_2_-induced oxidative stress models that are widely used to induce premature senescence and mimic hair loss conditions in vitro [22,35,36,37].

Finasteride, minoxidil, and baricitinib are the currently available FDA-approved drugs that prevent hair loss [38]. However, side effects such as a reduced libido have been reported [39]; therefore, the use of natural products is emerging as an alternative treatment for hair loss, and its importance is being revealed [40,41]. *Allium hookeri* (*A. hookeri*) is a species of the *Alliaceae* family and has been widely used as a medicinal plant in Asia for its bioactive compound contents [42]. Its previously reported biological activities have included anti-oxidative, anti-inflammatory, anti-adipogenic, and anti-microbial properties [42]. Furthermore, past studies have shown that the *A. hookeri* extract (AHE) had neuro-protective effects against oxidative damage and alleviated apoptosis in H_2_O_2_-treated PC12 cells [43]. In addition, AHE contains organosulfur compounds and was found to have significantly inhibited inflammatory cytokines in LPS-induced RAW264.7 cells via NF-κB down-regulation [44]. However, it remains unknown whether AHE exerts anti-oxidative, anti-senescent, or hair-inductive properties in HDPCs. Therefore, in this study, the biological effects of AHE on H_2_O_2_-induced cellular damage and intracellular signaling are investigated in HDPCs using an H_2_O_2_-stimulated oxidative stress model to mimic in vivo HF conditions. Additionally, this study is also aimed at proving the therapeutic potential of AHE for the treatment of hair loss via enhancing hair-inductive properties and improving oxidative stress-induced cellular damage in HDPCs.

## 2. Results

### 2.1. Allium hookeri Extract Increases Hair-Inductive Properties and Has No Cytotoxicity in HDPCs

*Allium* is a genus of monocotyledonous flowering plants and has hundreds of species. Previous studies have reported that several Allium species including *Allium tuberosum Rottler ex Spreng* and *Allium cepa* L. (also known as onion) have hair growth-promoting effects and can alleviate alopecia [45,46,47]. *Allium hookeri* is a wild herb within the genus *Allium* and various beneficial medical properties have been reported in vitro and in vivo experiments; however, there is no study on the effect of *A. hookeri* on viability and hair-inductive properties in HDPCs. Before evaluating the hair-inductive properties of AHE in HDPCs, its potential cytotoxicity was examined in HDPCs (Figure 1A–C). As shown in Figure 1A, there was no cytotoxic effect on 2D-cultured HDPCs treated with doses below 1000 µg/mL of AHE. HDPCs treated with 100, 200, and 400 µg/mL of AHE showed significant increases in cell viability, although no significant increase was observed at 1000 µg/mL. Thus, concentrations of 100, 200, and 400 µg/mL were selected for further experiments.

It has also been reported that 3D sphere culturing of DPCs can increase hair-inductive properties, with larger DP spheres exhibiting greater hair follicle inductivity [48,49]. In addition, Wnt/β-catenin signaling has been shown to be a major positive regulator inducing hair growth through the regulation of FGF and IGF signaling in DPCs [5]. The results shown in Figure 1B,C demonstrate that the size of HDP spheres was significantly increased at both 200 and 400 µg/mL. Subsequently, it was evaluated whether or not T-cell factor/lymphoid enhancer factor (TCF/LEF) transcriptional activity was increased in AHE-treated 293T cells. TOP-Flash luciferase reporter analysis also showed that AHE increased TCF/LEF-driven luciferase activity in a dose-dependent manner (Figure 1D).

Several previous studies have also identified that hair-inductive genes, including *LEF1*, *FGF7*, *WNT5A*, *IGF1*, *FGF2*, *VEGF*, *ALP*, and *VCAN*, are all essential factors that upregulate hair inductivity both in vitro and in vivo [3,50,51,52,53,54,55,56,57,58]. Therefore, here, it was evaluated whether or not AHE affected the expression of hair-inductive genes. HDPCs were treated with 400 µg/mL AHE for 8 h, whilst the mRNA levels of hair-inductive genes were evaluated through a quantitative reverse-transcription polymerase chain reaction (qRT-PCR). As shown in Figure 1E, AHE significantly increased the mRNA expression of hair-inductive genes *VCAN*, *VEGF*, *LEF1*, *FGF7*, *ALP*, *FGF2*, *IGF1*, and *WNT5A* compared to that in untreated control. Furthermore, the protein levels of IGF1, FGF2 and FGF7 in AHE-treated conditioned media were higher than the levels of those proteins in control-conditioned media (Figure 1F). In addition, the intracellular levels of those proteins in AHE-treated HDPCs were higher than were the levels of those proteins in the control (Figure 1G). These data suggested that AHE had no cytotoxicity and could effectively upregulate hair-inductive properties by increasing the sphere size and promoting the expression of hair-inductive genes in HDPCs.

### 2.2. A. hookeri Extract Has In Vitro Antioxidant Activities and Induces NRF2-Mediated Antioxidant Gene Expression

Previous studies have reported that excessive ROS generation and oxidative stress are implicated in the pathogenesis of alopecia [21,29,31,32,59]. Furthermore, DPCs from balding scalps were shown to be more sensitive to oxidative stress, whilst oxidative stress conditions induced hair growth inhibition via increasing the premature senescence of DPCs and secreting negative hair growth regulators [22,35]. Therefore, it was investigated whether or not AHE had an antioxidant effect in HDPCs. As shown in Figure 2A,B, AHE contained considerable polyphenol contents and showed radical scavenging activity. Additionally, to determine whether or not AHE reduced intracellular ROS in HDPCs, a DCF-DA assay was performed following cotreatment with indicated concentrations of AHE and H_2_O_2_. As a result, the intensity of DCF fluorescence significantly decreased in the presence of AHE compared to that in H_2_O_2_-treated HDPCs (Figure 2C,D). Considering that previous studies have reported that AHE possesses anti-oxidant capacities, we hypothesized that intracellular ROS scavenging activity is related to NRF2 signaling in HDPCs [60,61]. As shown in Figure 2E,F, it was proven that AHE increased the stabilization of NRF2 and the expression of ARE-responsive antioxidant genes (*HO1*, *NQO1*, *GSS*, *GPX1* and *SOD1*) in HDPCs in a dose-dependent manner. These results indicated that AHE had in vitro antioxidant activities and upregulated NRF2-induced anti-oxidant genes in HDPCs.

### 2.3. A. hookeri Extract Reduced H_2_O_2_-Induced Cytotoxicity and Apoptosis in HDPCs

Before analyzing the inhibitory effect of AHE on H_2_O_2_-stimulated cell dysfunction, the effect of H_2_O_2_ on HDPCs was evaluated. To establish the conditions for apoptosis or senescence, which are representative cellular responses to H_2_O_2_, the various concentrations of H_2_O_2_ (0–0.2 mM) were used for up to 3 d. As a result, in subsequent experiments, 0.1 or 0.2 mM H_2_O_2_ was used to induce senescence or apoptosis, respectively (Figure 3A). Since AHE had anti-oxidant effects in HDPCs, the ability of AHE to protect HDPCs from excessive ROS was measured via a WST-1 assay. Cells were pretreated with 0–400 µg/mL of AHE for either 12 h or 24 h before being stimulated with 0.2 mM H_2_O_2_. Cell viability was subsequently assessed after 24 h. As shown in Figure 3B,C, AHE significantly reduced cytotoxicity in a dose- and time-dependent manner. In the 24 h AHE pretreatment group, H_2_O_2_-treated cells showed a 42.08% reduction in cell viability compared to that in untreated cells; however, cells pretreated with 200 and 400 µg/mL AHE showed only 29.75% and 23.61% reductions compared to untreated cells, respectively. To confirm the inhibitory effect of AHE against H_2_O_2_-stimulated HDPCs, LDH leakage and crystal violet staining assays were performed. HDPCs were treated with AHE and H_2_O_2_ under the same conditions used for Figure 3B, and as shown in Figure 3D,E, it was confirmed that the cytotoxicity induced by H_2_O_2_ was alleviated with the AHE pretreatment in a dose-dependent manner.

Previous studies have also reported that apoptotic cells in HDPCs were observed in biopsies from alopecia areata patients, whilst hair follicle regression during catagen was characterized by apoptosis [62,63,64,65]. Therefore, herein, it is investigated whether or not apoptosis-related genes, *BCL2* and *BAX*, are involved in the alleviating effect of AHE on H_2_O_2_-induced apoptosis. As shown in Figure 3F,G, pretreatment with 200 and 400 µg/mL of AHE did not lead to an increase in mRNA expression, but increased the protein level of BCL2 compared to that in H_2_O_2_-treated HDPCs in a dose-dependent manner. In addition, levels of BAX, known as a pro-apoptotic factor, were decreased at both the mRNA and protein level in AHE-treated HDPCs compared to those in H_2_O_2_-treated HDPCs (Figure 3F,G). Therefore, these results indicated that AHE could alleviate H_2_O_2_-induced apoptosis via the regulation of apoptosis-related factors in HDPCs.

### 2.4. A. hookeri Extract Alleviates H_2_O_2_-Induced Cellular Senescence in HDPCs

Several previous studies reported that senescent DPCs induced the impairment of hair growth and that DPCs from balding scalps appeared to have more senescent phenotypes than non-balding scalps did [23,26,35]. Therefore, herein, it is determined whether or not AHE has anti-senescent effects and could prevent H_2_O_2_-induced senescence in HDPCs. After AHE and/or H_2_O_2_ treatment, a SA-β-gal assay was used to measure the amount of blue-stained, SA-β-gal-positive senescent cells under the same experimental conditions shown in Figure 4A. H_2_O_2_ (0.1 mM) treatment was found to have increased the percentage of senescent cells by 45.91% compared to that of untreated control cells; however, cells pretreated with 200 and 400 µg/mL of AHE decreased the percentage of senescent cells by 19.69% and 31.08% compared to that of H_2_O_2_-treated cells, respectively (Figure 4A,B). Subsequently, to confirm the mechanism via which AHE alleviated H_2_O_2_-induced senescence, senescence markers and SASP factors were analyzed through RT-PCR and qRT-PCR. As shown in Figure 4B,C, the mRNA expression of p21, a senescence marker and cell cycle dependent kinase inhibitor, was decreased in AHE-pretreated cells compared to that in H_2_O_2_-treated cells in a dose-dependent manner. Various SASP genes (*IL-1B*, *IL-6*, *IL-8*, and *CXCL1*) were also decreased in AHE-pretreated cells as opposed to H_2_O_2_-treated cells (Figure 4D,E).

These data ultimately suggested the possibility of AHE as a natural product with an anti-hair loss effect via the alleviation of H_2_O_2_-induced cellular senescence.

### 2.5. A. hookeri Extract Modulates β-Catenin, NRF2 and NF-κB Signaling Pathway in HDPCs

According to the previous results, AHE stimulated hair-inductive properties, increased ant-oxidative abilities, and reduced H_2_O_2_-induced cytotoxicity in HDPCs. Wnt/β-catenin signaling plays a major role in hair growth initiation, whilst NRF2 signaling is a master regulator of cellular resistance to oxidants [6,10,66,67,68]. In particular, it has been reported that NF-κB signaling is the master regulator stimulating the appearance of SASP [69,70,71]. Therefore, the molecular mechanism of AHE in cellular responses was investigated. As shown in Figure 5A, AHE decreased the phosphorylation of β-Catenin at serine 33/37 and threonine 41 residues whilst increasing the stabilization of β-Catenin protein levels in a dose-dependent manner. To determine the activity of β-Catenin, NRF2, and NF-κB, which are transcription factors, a cell fractionation assay was performed. Cell fractionation subsequently showed that β-Catenin, NRF2, and NF-κB translocation into the nucleus was modulated with AHE treatment in HDPCs (Figure 5B). The nuclear translocation level of β-Catenin and NRF2 increased in HDPCs treated with AHE; however, that of NF-κB (p50) decreased compared to that of PBS-treated controls. These results overall indicated the possibility that AHE regulated cellular responses by increasing β-Catenin and NRF2 signaling, whilst decreasing the transcription activity of NF-κB.

### 2.6. A. hookeri Extract Upregulates p38 and PKA Signaling Mediates the Inhibition of GSK3β Signaling Pathway in HDPCs

GSK3β is a major regulator of the β-Catenin, NRF2, and NF-κB signaling pathway [72,73,74]. Therefore, the effect of AHE on GSK3β activity and upstream proteins that regulated GSK3β was investigated. HDPCs were treated with 200 and 400 µg/mL of AHE for 24 h, with protein kinases related to the GSK3β signaling being assessed via Western blotting (Figure 6A,B). AHE treatment upregulated the phosphorylation of GSK3β at serine 9 and threonine 390 in a dose-dependent manner (Figure 6A). We then investigated upstream kinases that targeted the inhibitory phosphorylation site of GSK3β. The phosphorylation of AKT and ERK was not affected by AHE treatment. However, AHE treatment increased the phosphorylation of p38 at threonine 180 and tyrosine 182, and that of PKA at threonine 197 in a dose-dependent manner, respectively. These results suggested that AHE enhanced the activation of p38 and PKA signaling, followed by the upregulation of the inhibitory phosphorylation of GSK3β at the threonine 390 and serine 9 residue, respectively [75,76]. Collectively, these results suggested that the AHE-induced inhibition of GSK3β modulated the activation of β-Catenin and NRF2 signaling pathway and the inhibition of the NF-κB signaling pathway in HDPCs. The GSK3β-mediated signaling pathway upregulated the TCF/LEF-responsive hair-inductive genes and ARE-responsive antioxidant genes and could also alleviate cellular damage by H_2_O_2_ via inhibition of the NF-κB signaling pathway.

### 2.7. Characterization of A. hookeri Extract via High-Performance Liquid Chromatograph-High-Resolution Mass Spectrometry (HPLC-HRMS) Analysis

The various organic compounds found in AHE, which were detected through HPLC-HRMS, are shown in Table 1 and Figure 7. Alliin is a representative bioactive component found in AHE, as reported in a previous study [77]. Various amino acid analogues, L-(+)-leucine, DL-phenylalanine, and N-(4,5-Dihydro-1H-imidazol-2-yl)alanine, were detected in AHE. These results suggested that organic compounds in AHE were expected to have anti-oxidant and anti-inflammatory effects in HDPCs [78,79,80]. Using the data, we analyzed the effect of alliin on cell viability and the hair-inductive property in HDPCs. A WST-1-based cytotoxicity assay showed the non-toxic effect of alliin (concentration of up to 100 µM) on HDPCs (Appendix A). Then, we evaluated whether or not alliin affected the expression of hair-inductive genes including *VCAN*, *ALP* and *FGF7*. As shown in Appendix A, AHE significantly upregulated the mRNA expression of those genes compared to that in the untreated control. These data suggested that alliin is a representative bioactive component found in AHE, has no cytotoxicity and has a possible effect on the hair-inductive property via promoting the expression of hair-inductive genes in HDPCs.

## 3. Discussion

In spite of the previous studies showing that many *Allium* species promote hair growth and alleviate hair loss [45,46,47,81], the extraction method and specific *Allium* species used in this study are different from those used in previous studies. The extracts used in previous studies were generated from *Allium tuberosum Rolttler ex Spreng* and *Allium cepa* L., not from *Allium hookeri*, which was used in this study. Furthermore, ethanol and methanol extracting solvents and solvent-free juice preparation were used in previous studies rather than water as an extraction solvent, which was used in this study [45,46,47]. Previous studies have indicated that increased DP sphere sizes were associated with higher hair follicle inductivity [48,82,83]. In our results, it was demonstrated that AHE increased the sphere size of HDPCs in a dose-dependent manner, without causing cytotoxicity (Figure 1A–C). Furthermore, representative hair-inductive genes, including *LEF1*, *FGF7*, *WNT5A*, *IGF1*, *FGF2*, *VEGF*, *ALP*, and *VCAN*, have all been investigated as crucial elements that upregulate hair growth through DPCs [3,50,51,52,53,54,55,56,57,58]. In this study, it was also found that AHE significantly upregulated hair-inductive genes in HDPCs (Figure 1E). This work implied that by enlarging the sphere and stimulating the expression of hair-inductive genes in HDPCs, AHE could efficiently upregulate hair-inductive properties.

WNT/β-catenin signaling is significant in the commencement of hair growth [5,6,8,10]. Hence, several studies have used the stimulation of WNT/β-catenin signaling in DPCs as a therapeutic strategy for hair loss prevention [84,85,86]. In this study, it was found that AHE-treated HDPCs showed an increase in total β-catenin levels and in the nuclear translocation of β-catenin, alongside a decrease in β-catenin phosphorylation that induced the ubiquitination of β-catenin (Figure 5A,B). Additionally, it was revealed that AHE upregulated TCF/LEF-driven luciferase activity in a dose-dependent manner (Figure 1D). These results ultimately suggested that AHE stabilized β-catenin and activated the β-catenin signaling pathway in HDPCs.

Oxidative stress was shown to cause tissue dysfunction and was closely linked to skin aging and hair loss [87,88,89]. In particular, it was reported that DPCs under oxidative stress conditions can impair hair growth through the increase in negative hair growth regulators such as IL-6 [22,35]. Therefore, an attempt was made to clarify the antioxidant effects of AHE on HDPCs in this study. *A. hookeri* has previously been shown to have both anti-oxidant and anti-inflammatory properties [43,60,90]. As expected, AHE had different polyphenol contents and demonstrated radical scavenging abilities (Figure 2A,B). Furthermore, AHE reduced intracellular ROS levels in HDPCs via a DCF-DA assay (Figure 2C,D). Since NRF2 can regulate oxidative stress by combining with the antioxidant response element in the nucleus, we hypothesized that AHE upregulated NRF2 signaling in HDPCs [91]. It was subsequently demonstrated that AHE upregulated NRF2 stability and ARE-responsive antioxidant gene expression in HDPCs in a dose-dependent manner (Figure 2E,F). These findings overall showed that AHE upregulated NRF2-induced antioxidant genes in HDPCs and exhibited anti-oxidant properties in vitro.

Apoptosis and senescence are representative cellular responses to H_2_O_2_. Additionally, it has been reported that the apoptosis and senescence of DPCs is closely related to the pathogenesis of alopecia [22,26,62,92]. Furthermore, it was examined whether or not AHE, which has anti-oxidative properties, reduced apoptosis and senescence in H_2_O_2_-induced HDPCs. It was found that AHE alleviated the cytotoxicity in H_2_O_2_-induced HDPCs (Figure 3B,C). Furthermore, it was demonstrated in the LDH leakage assay that AHE decreased the LDH activity that had been increased by H_2_O_2_, whilst it was confirmed through the crystal violet staining assay that the pretreatment of AHE restored the H_2_O_2_-induced reduction in the proliferation rate (Figure 3D,E). To investigate the molecular mechanism of the anti-apoptotic effect of AHE, apoptosis-related factors such as BCL2 and BAX were analyzed via qRT-PCR or Western blotting. mRNA and protein levels of BAX were also significantly decreased in AHE-pretreated HDPCs compared to H_2_O_2_-treated HDPCs (Figure 3F,G). Similarly, pretreatment with AHE resulted in an increase in BCL2 protein levels in contrast to those in H_2_O_2_-treated HDPCs (Figure 3F,G). Collectively, our results suggest that AHE could control BAX and BCL2 in HDPCs to reduce H_2_O_2_-induced apoptosis.

In the present study, it was also proven that AHE had an anti-senescent effect on HDPCs under oxidative stress conditions. Aging of hair follicles causes tissue disorders such as hair thinning, hair loss, and miniaturization of the hair follicles [27,93,94]. In addition, oxidative stress in DPCs induces cellular senescence, which reduces the ability to induce hair follicle neogenesis [35]. In this work, various senescence markers, including SA-β-gal, p21, and p53, were reduced in AHE-pretreated HDPCs rather than in H_2_O_2_-treated HDPCs (Figure 4A–C). In particular, senescent cells secrete pro-inflammatory cytokines such as IL-6 and IL-8 called SASPs [95]. SASPs affect neighboring cells through cell-surface receptors, which then induce senescence surrounding cells through the upregulation of the NF-κB pathway [95,96]. As such, the expression level of SASPs was examined using qRT-PCR and RT-PCR, while various SAPS genes (*IL-1B*, *IL-6*, *IL-8*, and *CXCL1*), were found to be decreased in AHE-pretreated HDPCs as opposed to H_2_O_2_-treated HDPCs (Figure 4D,E). These data suggested the potential of AHE, a natural substance, to alleviate hair loss by reducing H_2_O_2_-induced cellular senescence.

GSK3β is an important kinase that regulates numerous substrates, including β-catenin, NRF2, and NF-κB [72,73,74]. GSK3β activity is regulated by its phosphorylation, which is mediated by various kinases, including AKT, PKA, ERK, and p38 [75]. Our results also showed that AHE enhanced the activation of p38 and PKA signaling, which induced the inhibitory phosphorylation of GSK3β at the threonine 390 residue and serine 9 residue, respectively (Figure 6A,B). However, phosphorylation of AKT and ERK was not affected by AHE treatment (Figure 6A,B). Overall, in this work, it was indicated that AHE stimulated hair inductive properties, increased antioxidative abilities, and reduced H_2_O_2_-induced cytotoxicity in HDPCs. Therefore, to investigate the molecular mechanism via which AHE acted on the cellular response, a cell fractionation assay was used to analyze the translocation levels of β-catenin, NRF2, and NF-κB, which are downstream targets of GSK3β. In HDPCs treated with AHE, the levels of nuclear translocation of β-catenin and NRF2 increased. However, the nuclear translocation level of NF-κB was decreased in AHE-treated HDPCs in comparison to that of controls treated with PBS (Figure 5A,B). These findings implied that the activity inhibition of GSK3β by AHE activated the signaling of β-catenin and NRF2, while inhibiting the activity of NF-κB in HDPCs. It was further found that AHE increased the phosphorylation levels of PKA and GSK3β; however, these effects were not shown in in HDPCs cotreated with AHE and a PKA inhibitor, daphnetin (Appendix A). This result suggested that AHE-induced phosphorylation of GSK3β is related to PKA activity in HDPCs. Further in-depth validations of the role of AHE on GSK3β phosphorylation are needed in further research.

In this study, it was found that AHE may support hair growth by enhancing hair-inductive properties and improving H_2_O_2_-induced cellular damage in cultured HDPCs. AHE enhanced hair growth properties in HDPCs by increasing β-catenin transcription activity, 3D sphere size, and hair-inductive genes. AHE also showed in vitro and cellular antioxidant properties by increasing the transcription level of NRF2. Under H_2_O_2_ conditions that induced the impairment of hair growth, AHE alleviated H_2_O_2_-induced cellular damage. AHE also increased the levels of the anti-apoptotic protein BCL2 while decreasing the levels of the pro-apoptotic factors of BAX. Furthermore, AHE also restored H_2_O_2_-induced cellular senescence in a dose-dependent manner, whilst significantly reducing the levels of CDKI and SASP genes in HDPCs. The therapeutic effects of AHE on alopecia and its definitive molecular mechanism still require further evaluation.

Furthermore, HPLC-HRMS analysis revealed that alliin is a representative bioactive component found in AHE. The alliin of AHE demonstrated the possibility of hair-inductive properties in a cell viability assay and expression analysis of hair-inductive genes including *VCAN*, *ALPL* and *FGF7*, but further validation is required to determine whether or not it can be a reliable component for hair -inductive properties in HDPCs and hair follicles.

In conclusion, this study provides us with the hair-inductive and antioxidant properties of AHE as a new cosmetic ingredient that can overcome the side effects of conventional alopecia treatment.

## 4. Materials and Methods

### 4.1. Cell Culture and Preparation of A. hookeri Extract

HDPCs provided by PromoCell (Heidelberg, Germany) were purchased and maintained in a follicle growth medium kit (Promocell, Heidelberg, Germany) with 5% CO_2_ and at 37 °C. Passage 4 to 6 HDPCs were used in in vitro cultivation and all the experiments in this study. *Allium hookeri* used in this study were collected in March 2022 from a cultivation area located in the Pyeongchang region (Gangwon-do, Republic of Korea). The plants were washed with distilled water three times and air-dried at room temperature (in the shade). Fifty grams of dried *A. hookeri* was chopped and ground into a powder using a grinder (SMX-3500GN, Shinil Industrial Co., Ltd., Seoul, Republic of Korea). Then, 50 g of the ground *A. hookeri* powder was mixed with 900 mL of distilled hot water (80 °C) for 4 h. The obtained extracts were filtered through Whatman filter paper No. 1 (Whatman, Maidstone, UK) followed by ultrafiltration through a sterile 0.2 µm bottle-top vacuum filter (Corning, Corning, NY, USA). Using HPLC-HRMS and DPPH analyses, the extract was determined to have the main component, alliin (MedChemExpress, Monmouth Junction, NJ, USA), and a radical scavenging effect; therefore, these two methods were used to monitor the quality of the extracts.

### 4.2. HPLC-HRMS Analysis

The extract was analyzed using HPLC-HRMS analysis. The chemical profiles of AHE were assessed using a Thermo Ultimate-3000 UPLC system (Thermo fisher scientific, Waltham, MA, USA) and an ACQUITY BEH C18 column (1.7 µm, 150 × 2.1 mm). The gradient conditions were as follows: 0 min (5% B), 0–5 min (5% B), 5–20 min (70% B), and 20–27 min (100% B). The flow rate was set to 0.4 mL/min, and the injection volume was 2 µL. The data were analyzed using Xcaliber software (Version 4.3, Thermo Finnigan, San Jose, CA, USA).

### 4.3. Total Polyphenol Contents

The Folin–Denis (Folin and Denis, 1912) method was modified and conducted [97]. A 2N Folin–Ciocalteau phenol reagent (Sigma-Aldrich, St. Louis, MO, USA) was used to determine the total polyphenol contents. Gallic acid (Sigma-Aldrich, St. Louis, MO, USA) was used to analyze the amount of TPC. AHE was mixed and reacted with the 2N Folin–Ciocalteau phenol reagent at different concentrations for 5 min. Then, 10% Na_2_CO_3_ (Sigma-Aldrich, St. Louis, MO, USA) was added. After incubation for 1 h in a dark room, total polyphenol contents were assessed via measuring absorbance at 765 nm using a microplate reader. The data were calculated as gallic acid equivalents (GAE) on a DW (GAE µg/g DW).

### 4.4. DPPH Radical Scavenging Activity

The method of Blois (1958) was modified to measure the radical scavenging activity of AHE [98]. DPPH (Sigma-Aldrich, St. Louis, MO, USA), a reagent containing free radicals, was used, and as a positive control, L-ascorbic acid (Sigma-Aldrich, St. Louis, MO, USA) was used for comparison. Each different concentration of AHE and a 0.2 mM DPPH reagent were combined and reacted for 10 min. Then, the absorbance was measured at 517 nm using a microplate reader.

### 4.5. Cell Viability Assay

To assess cell viability, HDPCs treated with AHE were assessed using the WST-1 assay. The HDPCs were plated on a 96-well plate and incubated at 37 °C for 24 h. Then, the HDPCs were treated with various concentrations of AHE for 48 h. Subsequently, the the EZ-Cytox mixture (100 µL/well) (DoGenBio, Seoul, Republic of Korea) was added to the cells and incubated with the cells for 30 min. Cell viability was analyzed using a microplate reader at 450 nm.

### 4.6. LDH Leakage Assay

The effect of AHE on the cytotoxicity of H_2_O_2_-induced HDPCs was assessed using the EZ-LDH Cell Cytotoxicity Assay Kit reagent (Biomax, Guri-si, Gyeonggi-do, Republic of Korea) following the manufacturer’s protocols. The lactate dehydrogenase released from damaged and dead cells was assessed from the culture medium. The HDPCs were seeded in a 96-well plate at 4 × 10^3^ cells per well. After overnight growth, the HDPCs were treated with different concentrations of AHE for 24 h. Subsequently, the cells were incubated with 0.2 mM H_2_O_2_ for 24 h. Afterward, 10 µL of each of the supernatant was dispensed into a new 96-well plate, and 100 µL of each of the mixture of the EZ-LDH Cell Cytotoxicity Assay Kit reagent was added and reacted for 30 min. Cytotoxicity was analyzed using a microplate reader at 450 nm.

### 4.7. Crystal Violet Staining

A crystal violet staining assay was examined as described by Maria Feoktistova et al. [99]. After seeding 1 × 10^5^ cells per well in a 6-well plate, the HDPCs were treated with different concentrations of AHE for 24 h. Subsequently, the cells were incubated with 0.2 mM H_2_O_2_ for 24 h. Attached viable cells were stained by a 0.5% crystal violet (Biopure, Havant, UK) solution for 60 min with gentle shaking. Then, the stained cells were washed 10 times with distilled water and completely dried. To dissolve crystal violet dye that stained the attached cells, a 1% sodium dodecyl sulfate (SDS) (Biopure, Havant, UK) solution was used for extraction. For colorimetric analysis, the absorbance was determined at 570 nm using a microplate reader.

### 4.8. Intracellular ROS Measurement

Th cellular ROS level was measured using H_2_DCFDA (Sigma-Aldrich, St. Louis, MO, USA) and a previously modified method [100]. Briefly, HDPCs growing in fluorescence microtiter 96-well plates were loaded with 10 µM DCF-DA in PBS and incubated for 30 min in a dark room. Then, to induce intracellular ROS generation, the cells were treated with 1 mM H_2_O_2_ and incubated for 10 min in a dark room. ROS generation was determined by measuring dichlorofluorescein (DCF) using a fluorescence microplate reader at excitation wavelengths of 485 nm and emission wavelengths of 520 nm.

### 4.9. 3D Spheroid Culture of HDPCs

A spheroid culture was performed as previously described [84]. A total of 2 × 10^3^ HDPCs were plated on 96-well round-bottom ultra-low-attachment microplates (Corning, Glendale, AZ, USA) and were incubated for 24 h. Then, spheroids were treated with different concentrations of AHE and further incubated for 24 h and 48 h. The diameter of the spheroids was measured using phase-contrast microscopy.

### 4.10. TOPFlash Luciferase Reporter Assay

For the TOPFlash luciferase reporter assay, 293T cells were plated on a 6-well plate at 3 × 10^5^ cells per well and maintained at 37 °C for 24 h. Subsequently, the cells were co-transfected with a pSV-β-galactosidase (pSV-β-gal) plasmid and TCF/LEF response element-driven luciferase reporter plasmid using the Lipofectamine reagent (Thermo fisher scientific, Walthan, MA, USA) Then, the cells were treated with indicated concentrations of AHH for 24 h. Subsequently, transfected cells were lysed using a 5× passive lysis buffer (Promega, Madison, WI, USA). The cell lysates were reacted with d-luciferin (Sigma-Aldrich, St. Louis, MO, USA) and luciferase activity was measured using a microplate reader. Beta-galactosidase activity was assessed using Beta-Glo Assay System (Promega, Madison, WI, USA). Relative luciferase activity was analyzed by normalizing luciferase activity to β-galactosidase activity.

### 4.11. SA-β-gal-Based Cellular Senescence Analysis

Cellular senescence level was assessed using a SA-β-gal staining assay. Briefly, HDPCs were seeded in a 12-well plate at 2 × 10^4^ cells per well and were incubated at 37 °C for 24 h. Then, the HDPCs were pretreated with various concentrations of AHE for 24 h. Subsequently, the cells were incubated with 0.1 mM H_2_O_2_ for 72 h. The treated cells were fixed via the addition of 4% formaldehyde for 15 min. After fixation, a SA-β-gal staining solution (Cell Signaling Technology, Danvers, MA, USA) was added to the cells and reacted at 37 °C for 48 h. Senescent cells were observed and counted using a bright-field microscope, and the percentages were analyzed.

### 4.12. PCR and Quantitative Real-Time PCR

HDPCs were seeded in 100 mm dishes and were incubated overnight. Then, the cells were treated with different concentrations of AHE. Total RNA was extracted using the RiboEx reagent (Geneall Biotechnology, Seoul, Republic of Korea) in accordance with the manufacturer’s instructions. The complementary DNA was synthesized using 1 µg of total RNA, 2.5 mM dNTPs, oligo dT primers, 0.1 M DTT, 5X Firse-Strand Buffer, and M-MLV reverse transcriptase (Thermo fisher scientific, Waltham, MA, USA). *GAPDH* was used to normalize the gene expression level. The primer sequence of the specific gene used in the analysis is as follows. *WNT5A*, 5′-TTGAAGCCAATTCTTGGTGGTCGC-3′ (forward) and 5′-TGGTCCTGATACAAGTGGCACAGT-3′ (reverse); *ALP*, 5′-CAAACCGAGATACAAGCACTCCC-3′ (forward) and 5′-CGAAGAGACCCAATAGGT AGTCCAC-3′ (reverse); *VCAN*, 5′-GGCAATCTATTTTACCAGGACCTGAT-3′ (forward) and 5′-TGGCACACAGGTGCATACGT-3′ (reverse); *FGF2*, 5′-ACCTGCAGACTGCTTTTTGCC-3′ (forward) and 5′-GGTGCCACGTGAGAGCAGAGC-3′ (reverse); *VEGF*, 5′-GAGGGCAGAATCATCACGAAGT-3′ (forward) and 5′-CACCAGGGTCTCGATTGGAT-3′ (reverse); *LEF1* 5′-CCTGGTCCCCAC ACAACT-3′ (forward) and 5′-GGCTCCTGCTCCTTTCTCTG-3′ (reverse); *IGF1*, 5′-CTCTTCTACCTGGCGCTGTG-3′ (forward) and 5′-CATACCCTGTGGGCTTGTTG-3′ (reverse); *FGF7*, 5′-TCTGTCGAACACAGTGGTACCTGAG-3′ (forward) and 5′-GCCACTGTCCTGATTTCCATGA-3′ (reverse); *p16*, 5′-TGCCTTTTCACTGTGTTGGA-3′ (forward) and 5′-GCCATTTGCTAGCAGTGTGA-3′ (reverse); *p21*, 5′-GAACTTCGACTTTGTCACCGAGAC-3′ (forward) and 5′-TGGAGTGGTAGAAATCTGTCATGCT-3′ (reverse); *p53*, 5′-CCAGGGCAGCTACGGTTTC-3′ (forward) and 5′-CTCCGTCATGTGCTGTGACTG-3′ (reverse); *HO1*, 5′-GCCCTTCAGCATCCTCAGTTCC-3′ (forward) and 5′-AGTGGTCAT GGCCGTGTCAAC-3′ (reverse); *NQO1*, 5′-GGGAGACAGCCTCTTACTTGCC-3′ (forward) and 5′-AACACCCAGCCGTCAGCTATTG-3′ (reverse); *GSS*, 5′-TAGATGCCCCACGTGCTTGT-3′ (forward) and 5′-ATCCTCATGGAGAAGATCGA-3′ (reverse); *SOD1*, 5′-CCAGTGCAGGGCATCATCA-3′ (forward) and 5′-TTGGCCCACCGTGTTTTCT-3′ (reverse); *GPX1*, 5′-GCAGCTCGTTCATCTGGGTG-3′ (forward) and 5′-ATGTGTGCTGCTCGGCTAGC-3′ (reverse); *IL-1β*, 5′-TTCCCTGCCCACAGACCTTCC-3′ (forward) and 5′-TGCATCGTGCACATAAGCCTCG-3′ (reverse); *IL-6*, 5′-GTAGCCGCCCCACACAGA-3′ (forward) and 5′-CATGTCTCCTTTCTCAGGGCTG-3′ (reverse); *IL-8*, 5′-TCTCTTGGCAGCCTTCCTGA-3′ (forward) and 5′-TTCTGTGTTGGCGCAGTGTG-3′ (reverse); *CXCL1*, 5′-AGGCCACCTGGATTGTGCCTAA-3′ (forward) and 5′-GCATGTTGCAGGCTCCTCAGAA-3′ (reverse); *Bax*, 5′-CCCGAGAGGTCTTTTTCCGAG-3′ (forward) and 5′-CCAGCCCATGATGGTTCTGAT-3′ (reverse); *BCL2*, 5′-CATGTGTGTGGAGAGCGTCAAC-3′ (forward) and 5′-CAGATAGGCACCCAGGGTGAT-3′ (reverse); and GAPDH, 5′-TCCAAAATCAAGTGGGGCGATGC-3′ (forward) and 5′-GCCAGTAGAGGCAGGGATGATGT-3′ (reverse).

### 4.13. Western Blot Analysis

HDPCs were seeded in 100 mm dishes at 5 × 10^5^ cells each and cultured for 24 h. AHE was treated at indicated concentrations for 24 h. Then, the cells were lysed and total cell lysates were prepared. Intracellular (nuclear and cytoplasmic) fractions were separated using NE-PER reagents (Thermo fisher scientific, Waltham, MA, USA). The 20 µg protein samples were examined via Western blotting with the corresponding antibodies: β-catenin (1:1000, Cell Signaling Technology, Danvers, MA, USA), phospho-β-catenin (Ser33/Ser39/Thr41) (1:1000, CST), GSK3β (1:1000, CST), phospho-GSK3β (Ser9) (1:1000, CST), phospho-GSK3β (Thr390) (1:1000, CST), AKT (1:1000, CST), phospho-AKT (ser473) (1:1000, CST), PKA (1:1000, CST), phospho-PKA (Thr198) (1:1000, CST), ERK (1:1000, CST), phospho-ERK (Thr202/Tyr204) (1:1000, CST), BCL2 (1:1000, CST), BAX (1:1000, CST), p16 (1:1000, CST), p21 (1:1000, CST), NRF2 (1:1000, CST), p53 (1:200, Santa Cruz, Dallas, TX, USA), NF-κB p50 (1:200, Santa Cruz), VEGF (1:200, Santa Cruz), IGF1 (1:200, Santa Cruz), FGF2 (1:200, Santa Cruz), FGF7 (1:200, Santa Cruz), and β-actin (1:1000, Santa Cruz). Daphnetin, a potent inhibitor of protein kinase A, was purchased from Sigma-Aldrich (St. Louis, MO, USA). The Western blot was analyzed using a chemiluminescence detector.

### 4.14. Determination of Secreted Proteins

An equal aliquot of the conditioned culture media from an equal number of HDPCs was used to determine the total secreted proteins released into the culture media. Ice-cold acetone was mixed with the culture media in a ratio of 4:1and incubated at −20 °C for 1 h. The protein pellet was precipitated following centrifugation at 15,000× *g* for 15 min at 4 °C. After washing with 80% ice-cold acetone, the pellets were resuspended in a SDS-PAGE sample buffer and subjected to Western blot analysis.

### 4.15. Statistical Analysis

One-way analysis of variance (ANOVA) was used to assess the statistical significance of the differences among the treatment groups. For each statistically significant treatment effect, Tukey’s test was used for comparisons between multiple group means. The data are expressed as mean ± standard deviation (SD). Differences of *p* < 0.05 were considered statistically significant.

## Figures and Tables

**Figure 1 plants-12-01919-f001:**
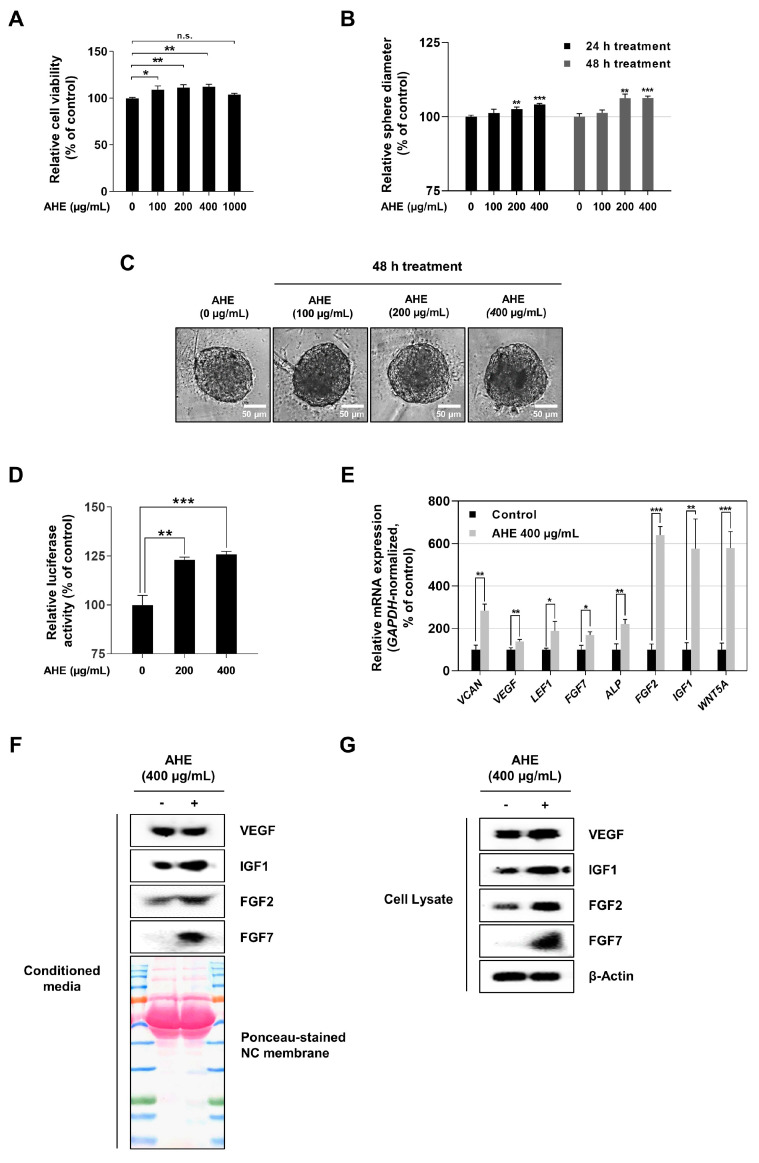
Effects of *A. hookeri* extract on cell viability and hair-inductive properties in HDPCs. (**A**) HDPCs treated with the indicated concentrations of AHE for 48 h. Cell viability of HDPCs was measured via WST-1 assay. (**B**) Use of a 96-well ultra-low attachment microplate to measure the diameter of HDP spheres. Spheroids were incubated with 100, 200, and 400 µg/mL of AHE for 24 h and 48 h. (**C**) Representative images of AHE-treated HDP spheres for 48 h captured by phase-contrast microscopy. (**D**) 293T cells treated with 200 and 400 µg/mL AHE for 24 h. TOP-Flash luciferase reporter assay was assessed using luminescence and normalized against β-galactosidase activity. (**E**) mRNA level of hair-inductive genes (*VCAN*, *VEGF*, *LEF1*, *FGF7*, *ALP*, *FGF2*, *WNT5A*, and *IGF1*) assessed via qRT-PCR and then normalized against *GAPDH*. (**F**) Protein levels of VEGF, IGF1, FGF2 and FGF7 in AHE-treated conditioned media determined by Western blotting and Ponceaus staining of the blot membrane used as the loading control. (**G**) Intracellular proteins of VEGF, IGF1, FGF2 and FGF7 analyzed by Western blotting with β-actin serving as a loading control. The results are presented as the mean ± SD of three independent experiments and were analyzed with one-way ANOVA analysis followed by Tukey’s test. AHE, *Allium hookeri* extract; HDPCs, human dermal papilla cells; qRT-PCR, quantitative reverse-transcription polymerase chain reaction. * *p* < 0.05, ** *p* < 0.01, and *** *p* < 0.001.

**Figure 2 plants-12-01919-f002:**
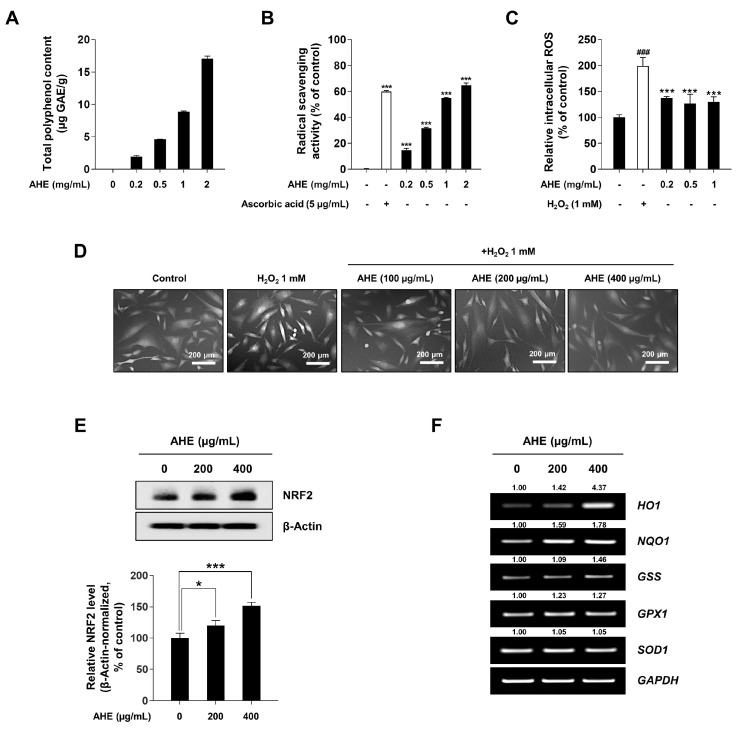
Antioxidant activities and induction of antioxidant-related genes by *A. hookeri* extract in HDPCs. (**A**) Total polyphenol content of AHE. The standard curve of 0–100 μg/mL gallic acid was y = 0.0324x + 0.0545; R^2^ = 0.9989. (**B**) DPPH radical scavenging activity of AHE. Ascorbic acid (5 µg/mL) was used as a positive control. (**C**) Intracellular ROS level assessed by a DCF-DA microplate reader assay. (**D**) Representative UV microscope Axiovert 200-captured images of DCF fluorescence. (**E**) HDPCs incubated with indicated concentrations of AHE for 24 h. NRF2 stabilization was analyzed by Western blotting whilst β-actin served as a loading control. (**F**) mRNA level of antioxidant-related genes (*HO1*, *NQO1*, *GSS*, *GPX1*, and *SOD1*) detected via RT-PCR, with *GAPDH* serving as a loading control. The results are presented as the mean ± SD of three independent experiments and analyzed with one-way ANOVA analysis followed by Tukey’s test. AHE, *Allium hookeri* extract; DPPH, 2,2-diphenyl-1-picrylhydrazyl; ROS, reactive oxygen species; DCF-DA, 2,7-dichlorofluoroscin diacetate; HDPCs, human dermal papilla cells; RT-PCR, reverse-transcription polymerase chain reaction. * *p* < 0.05; ^###^, *** *p* < 0.001.

**Figure 3 plants-12-01919-f003:**
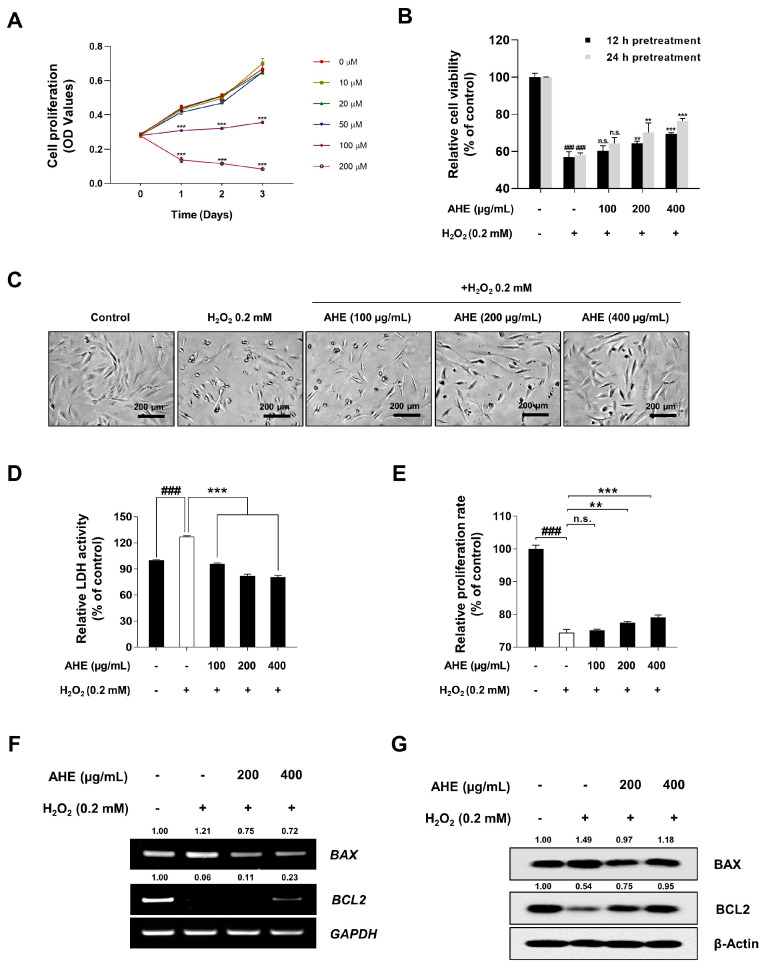
Protective effects of *A. hookeri* extract on H_2_O_2_-induced apoptosis in HDPCs. (**A**) HDPCs incubated with various concentrations of H_2_O_2_ for indicated times. Cell proliferation was determined via WST-1 assay. Percentage of live cells, assessed by cell viability assay with WST-1, following H_2_O_2_ treatment time selected for subsequent analysis. (**B**) Reduced cytotoxicity of HDPCs treated with AHE against H_2_O_2_-stimulation. HDPCs were incubated with AHE for 12 h and 24 h followed by treatment with 0.2 mM H_2_O_2_ for 24 h, with cytotoxicity being measured through a WST-1 assay. (**C**) Images of H_2_O_2_-induced HDPCs captured by phase-contrast microscopy. (**D**) Oxidative stress-induced cytotoxicity evaluated via LDH leakage assay. (**E**) Proliferation rate of H_2_O_2_-induced HDPCs measured via crystal violet assay. (**F**) The mRNA expression level of apoptosis-related genes (*BAX* and *BCL2*) was detected via RT-PCR, and *GAPDH* served as a loading control. (**G**) Protein level of BAX and BCL2 analyzed via Western blotting whilst β-actin served as a loading control. The results are presented as the mean ± SD of three independent experiments and analyzed with one-way ANOVA analysis followed by Tukey’s test. AHE, *Allium hookeri* extract; HDPCs, human dermal papilla cells; LDH, lactate dehydrogenase; RT-PCR, reverse-transcription polymerase chain reaction; qRT-PCR, quantitative reverse-transcription polymerase chain reaction. ** *p* < 0.01; ^###^, *** *p* < 0.001.

**Figure 4 plants-12-01919-f004:**
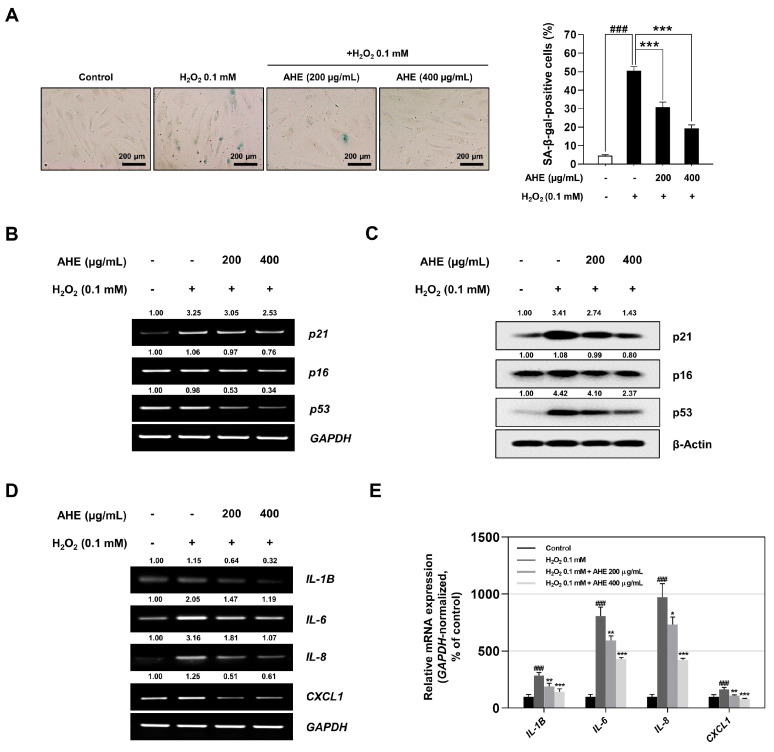
Protective effects of *A. hookeri* extract on H_2_O_2_-induced senescence in HDPCs. (**A**) SA-β-gal staining. Prior to being treated with 0.1 mM H_2_O_2_ for 72 h, HDPCs were pretreated with AHE for 24 h. SA-β-gal positive cells were then observed under a microscope (scale bars, 200 μm). (**B**) The expression level of senescence-associated genes (*p16*, *p21*, and *p53*) in HDPCs cotreated with AHE and 0.1 mM H_2_O_2_ assessed via RT-PCR and normalized against *GAPDH*. (**C**) Protein levels of p16, p21, and p53 analyzed via Western blotting with β-actin serving as a loading control. (**D**) Expression level of senescence associated secretory phenotype (SASP) genes in HDPCs determined via RT-PCR. (**E**) mRNA levels of *IL1β*, *IL6*, *IL8*, and *CXCL1* assessed via qRT-PCR and normalized against *GAPDH*. The results are presented as the mean ± SD of three independent experiments and analyzed with one-way ANOVA analysis followed by Tukey’s test. AHE, *Allium hookeri* extract; HDPCs, human dermal papilla cells; RT-PCR, reverse-transcription polymerase chain reaction; qRT-PCR, quantitative reverse-transcription polymerase chain reaction. * *p* < 0.05; ** *p* < 0.01; ^###^, *** *p* < 0.001.

**Figure 5 plants-12-01919-f005:**
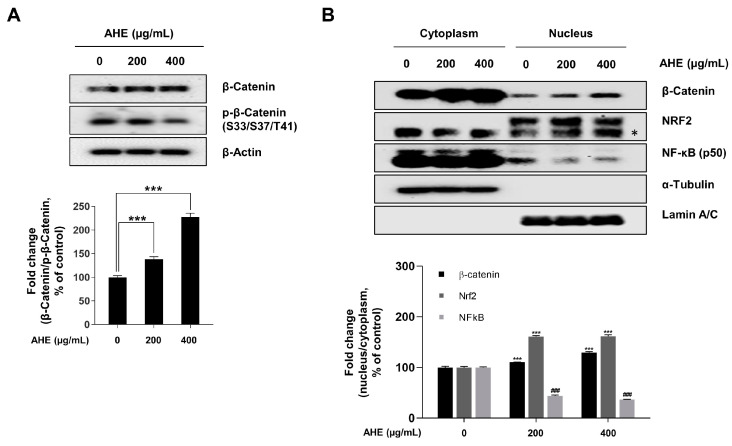
Effect of *A. hookeri* extract on β-Catenin, NRF2, and NF-κB signaling pathway in HDPCs. (**A**) β-Catenin stabilization assessed via Western blotting with β-Actin serving as a loading control. (**B**) Nuclear translocation of β-Catenin, NRF2, and NF-κB assessed via Western blotting. α-Tubulin and Lamin A/C served as loading controls for the cytoplasmic and nuclear fractions, respectively. Quantification of the protein level was carried out using ImageJ. The results are presented as the mean ± SD of three independent experiments and analyzed with one-way ANOVA analysis followed by Tukey’s test. AHE, *Allium hookeri* extract; HDPCs, human dermal papilla cells. ^###^, *** *p* < 0.001. * indicates NRF2 target band.

**Figure 6 plants-12-01919-f006:**
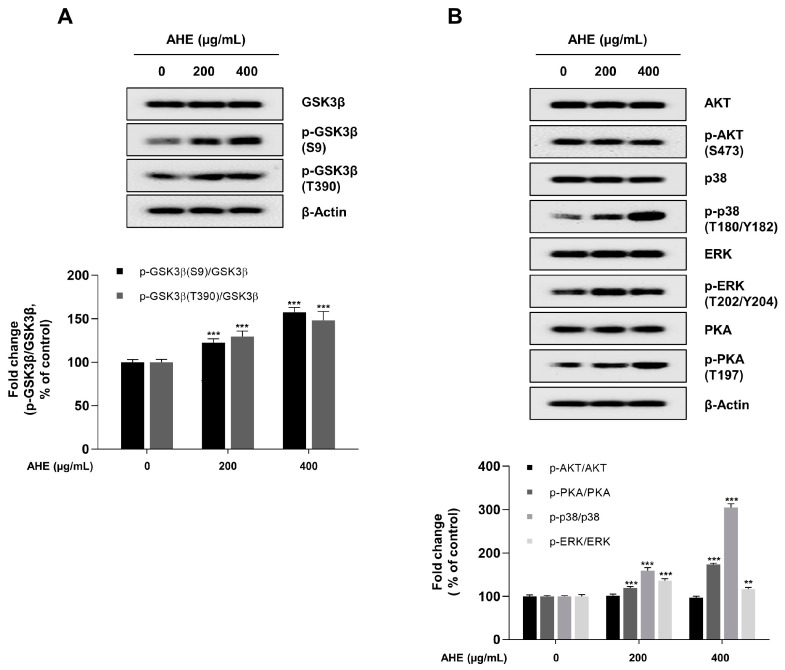
Effect of *A. hookeri* extract on the inhibitory phosphorylation of GSK3β via PKA and p38 signaling pathways. (**A**) Inhibitory phosphorylation of GSK3β analyzed via Western blotting with β-actin serving as a loading control. (**B**) Protein levels of GSK3β upstream signaling target genes assessed via Western blotting with β-actin serving as a loading control. Quantification of protein level was carried out using the ImageJ software. The results are presented as the mean ± SD of three independent experiments and analyzed with one-way ANOVA analysis followed by Tukey’s test. AHE, *Allium hookeri* extract; HDPCs, human dermal papilla cells. ** *p* < 0.01; *** *p* < 0.001.

**Figure 7 plants-12-01919-f007:**
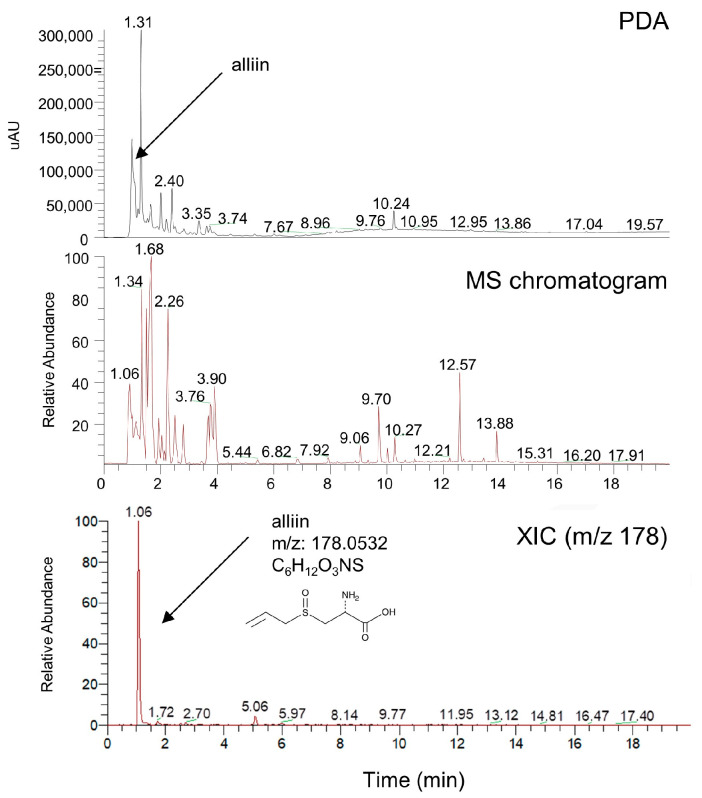
Characterization of compounds from *A. hookeri* extract with HPLC-HRMS analysis.

**Table 1 plants-12-01919-t001:** HR-MS data of identified molecules in the *A. hookeri* extract.

Number	Name	Formula	M.W	RT (min)
1	Alliin	C_6_H_11_O_3_NS	177.22	1.06
2	N-(4,5-Dihydro-1H-imidazol-2-yl)alanine	C_6_H_11_N_3_	157.170	1.34
3	L-(+)-Leucine	C_6_H_13_NO_2_	131.173	1.68
4	DL-Phenylalanine	C_9_H_11_NO_2_	165.189	2.26

## Data Availability

Not applicable.

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
