# Peer review of "Effects of Allium hookeri Extracts on Hair-Inductive and Anti-Oxidative Properties in Human Dermal Papilla Cells"

_plants, 2023, doi:10.3390/plants12091919_

Round 1
Reviewer 1 Report
Park et al reported the effect of AHE on the hair-inductive features of DPCs. They also tested the anti-oxidative damage effect of AHE on DPCs and proposed a potential mechanism. The work is of interest. However, some concerns arose when reviewing the manuscript.
1. HDPCs lost hair-inductive properties during the passage. Thus, the times of passage will impact the results. How many times did you passage the cells? Are all the experiments using HDPCs at the same passage time?
2. Figure 1, the author tested the diameter of spheres and the mRNA expression of some molecules, and claimed that the hair-inductive properties were changed. It is far from enough. The secretion of hair-inductive molecules is a main prosperity of DPCs. They should also test the protein expression and secretion of these molecules. In addition, the photos of the cell spheres should be shown.
3. Figure 2 and Table 1, the authors identified some an-oxidative damage molecules. It is better to put the data to the end of the result. In addition, they should compare the effect of these molecules on cell viability and hair-inductive properties in HDPCs. The manuscript lacks novelty. In this way, they can provide new clues for future research.
4. Figure 7, the authors claim that AHE inhibits the phosphorylation of GSK3β via PKA and p38 signaling pathways. However, only expression data were provided. The gain of function and loss of function experiments are needed to support their claim.
5. Page 3, “Previous studies have reported that several Allium species have hair growth-promoting effects and can alleviate alopecia. However, there is no scientific evidence yet for Allium hookeri in terms of hair growth-promoting properties.” What do you want to express? It seems that the two sentences are contradictory.
Moderate editing of the English language is needed.
Reviewer 2 Report
Comments:
1. In my opinion, the title of the article should be changed. The Authors used a statement when proposing the topic, and only the results in the paper allow the conclusion that the extract enhances and improves properties. A different title that does not state the results obtained in the paper should be considered.
2. Throughout the paper the Authors use the statements 'we investigated', 'we found' etc. Authors should use the passive voice throughout the manuscript. Concerns the lines: 12, 13, 90,93, 103, 114, 120, 160, 163, 189, 201, 208, 221, 223, 275, 299, 329, 336, 342, 352, 359, 365, 367, 379, 390, 409.
3. In the text, avoid phrases such as "our results" , "our findings". They can be replaced by: „in this work, in this article etc. Concerns the lines: 270, 337, 383, 399.
4. Noteworthy is the impressive literature (perhaps even too extensive for the purposes of this manuscript) – 102 references. Well done!
After taking into account all comments, I recommend the article for publication in the journal Plants, after minor revision.
Reviewer 3 Report
This study shows that an extract from Allium hookeri can increase many cellular markers of hair growth. Comments:
1. There is no description of the source of the extract - just a place name. No one could reproduce this work from this information. The authors need to explain where the material came from, how it was processed, and what quality control testing was done to insure the material was the same in all the tests.
2. The method is hot water extraction of the Allium. However, references 45 & 47 supporting hair growth used ethanol, methanol or butanol extraction, not water. The authors should note in the text that the papers supporting their work used a different method of extraction.
3. There is no explanation of the abbreviation "TCF/LEF" (line 114)
4. The authors should change "improved cytotoxicity" to "reduced cytotoxicity" (line 271 and elsewhere)
Round 2
Reviewer 1 Report
The authors revised the manuscript accordingly.